# Federated Model Heterogeneous Matryoshka Representation Learning

**Liping Yi**[1,2], **Han Yu**[2], **Chao Ren**[2], **Gang Wang**[1,*], **Xiaoguang Liu**[1,*], **Xiaoxiao Li**[3,4]

[1]College of Computer Science, TMCC, SysNet, DISSec, GTIISC, Nankai University, China
[2]College of Computing and Data Science, Nanyang Technological University, Singapore
[3]Department of Electrical and Computer Engineering, The University of British Columbia, Canada
[4]Vector Institute, Canada

`{yiliping, wgzwp, liuxg}@nbjl.nankai.edu.cn`
`{han.yu, chao.ren}@ntu.edu.sg, xiaoxiao.li@ece.ubc.ca`

## Abstract

Model heterogeneous federated learning (MHeteroFL) enables FL clients to collaboratively train models with heterogeneous structures in a distributed fashion. However, existing MHeteroFL methods rely on training loss to transfer knowledge between the client model and the server model, resulting in limited knowledge exchange. To address this limitation, we propose the Federated model heterogeneous Matryoshka Representation Learning (`FedMRL`) approach for supervised learning tasks. It adds an auxiliary small homogeneous model shared by clients with heterogeneous local models. (1) The generalized and personalized representations extracted by the two models' feature extractors are fused by a personalized lightweight representation projector. This step enables representation fusion to adapt to local data distribution. (2) The fused representation is then used to construct Matryoshka representations with multi-dimensional and multi-granular embedded representations learned by the global homogeneous model header and the local heterogeneous model header. This step facilitates multi-perspective representation learning and improves model learning capability. Theoretical analysis shows that `FedMRL` achieves a $\mathcal{O}(1/T)$ non-convex convergence rate. Extensive experiments on benchmark datasets demonstrate its superior model accuracy with low communication and computational costs compared to seven state-of-the-art baselines. It achieves up to $8.48\%$ and $24.94\%$ accuracy improvement compared with the state-of-the-art and the best same-category baseline, respectively.

## 1 Introduction

Traditional federated learning (FL) [32, 47, 46, 12] often relies on a central FL server to coordinate multiple data owners (a.k.a., FL clients) to train a global shared model without exposing local data. In each communication round, the server broadcasts the global model to the clients. A client trains it on its local data and sends the updated local model to the FL server. The server aggregates local models to produce a new global model. These steps are repeated until the global model converges. During the runtime of FL, only model parameters are transmitted between the server and clients, preserving data privacy[14, 56, 51].

However, the above design cannot handle the following heterogeneity challenges [53] commonly found in practical FL applications: (1) Data heterogeneity [42]: FL clients' local data often follow non-independent and identically distributions (non-IID). A single global model produced by aggregating

---

*Corresponding authors

38th Conference on Neural Information Processing Systems (NeurIPS 2024).

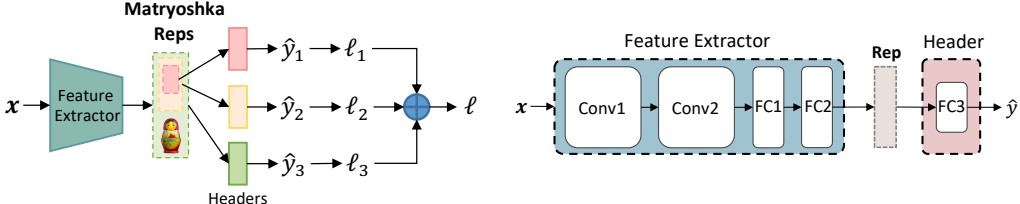

Figure 1: Left: Matryoshka Representation Learning. Right: Feature extractor and prediction header.

local models trained on non-IID data might not perform well on all clients [49, 48]. (2) System heterogeneity [11]: FL clients can have diverse system configurations in terms of computing power and network bandwidth. Training the same model structure among such clients means that the global model size must accommodate the weakest device, leading to sub-optimal performance on other more powerful clients [52, 54, 50]. (3) Model heterogeneity [43]: When FL clients are enterprises, they might have heterogeneous proprietary models which cannot be directly shared with others during FL training due to intellectual property (IP) protection concerns.

To address these challenges, the field of model heterogeneous federated learning (MHeteroFL) [55] has emerged. It enables FL clients to train local models with tailored structures suitable for local system resources and local data distributions. Existing MHeteroFL methods [41, 45] are limited in terms of knowledge transfer capabilities as they commonly leverage the training loss between server and client models for this purpose. This design leads to model performance bottlenecks, incurs high communication and computation costs, and risks exposing private local model structures and data.

Recently, Matryoshka Representation Learning (MRL) [24] has emerged to tailor representation dimensions based on the computational and storage costs required by downstream tasks to achieve a near-optimal trade-off between model performance and inference costs. As shown in Figure 1(left), the representation extracted by the feature extractor is constructed to form Matryoshka Representations involving a series of embedded representations ranging from low-to-high dimensions and coarse-to-fine granularities. Each of them is processed by a single output layer for calculating loss, and the sum of losses from all branches is used to update model parameters. This design is inspired by the insight that people often first perceive the coarse aspect of a target before observing the details, with multi-perspective observations enhancing understanding.

Inspired by MRL, we address the aforementioned limitations of MHeteroFL by proposing the Federated model heterogeneous Matryoshka Representation Learning (FedMRL) approach for supervised learning tasks. For each client, a shared global auxiliary homogeneous small model is added to interact with its heterogeneous local model. Both two models consist of a feature extractor and a prediction header, as depicted in Figure 1(right). FedMRL has two key design innovations. **(1) Adaptive Representation Fusion**: for each local data sample, the feature extractors of the two local models extract generalized and personalized representations, respectively. The two representations are spliced and then mapped to a fused representation by a lightweight personalized representation projector adapting to local non-IID data. **(2) Multi-Granularity Representation Learning**: the fused representation is used to construct Matryoshka Representations involving multi-dimension and multi-granularity embedded representations, which are processed by the prediction headers of the two models, respectively. The sum of their losses is used to update all models, which enhances the model learning capability owing to multi-perspective representation learning.

The personalized multi-granularity MRL enhances representation knowledge interaction between the homogeneous global model and the heterogeneous client local model. Each client's local model and data are not exposed during training for privacy-preservation. The server and clients only transmit the small homogeneous models, thereby incurring low communication costs. Each client only trains a small homogeneous model and a lightweight representation projector in addition, incurring low extra computational costs. We theoretically derive the $\mathcal{O}(1/T)$ non-convex convergence rate of FedMRL and verify that it can converge over time. Experiments on benchmark datasets comparing FedMRL against seven state-of-the-art baselines demonstrate its superiority. It improves model accuracy by up to $8.48\%$ and $24.94\%$ over the best baseline and the best same-category baseline, while incurring lower communication and computation costs.

## 2 Related Work

Existing MHeteroFL works can be divided into the following four categories.

**MHeteroFL with Adaptive Subnets.** These methods [3, 4, 5, 11, 16, 57, 65] construct heterogeneous local subnets of the global model by parameter pruning or special designs to match with each client's local system resources. The server aggregates heterogeneous local subnets wise parameters to generate a new global model. In cases where clients hold black-box local models with heterogeneous structures not derived from a common global model, the server is unable to aggregate them.

**MHeteroFL with Knowledge Distillation.** These methods [6, 8, 9, 17, 18, 19, 25, 26, 28, 30, 33, 35, 38, 39, 44, 58, 60] often perform knowledge distillation on heterogeneous client models by leveraging a public dataset with the same data distribution as the learning task. In practice, such a suitable public dataset can be hard to find. Others [13, 61, 62, 64] train a generator to synthesize a shared dataset to deal with this issue. However, this incurs high training costs. The rest (FD [21], `FedProto` [43] and others [1, 2, 15, 53, 59]) share the intermediate information of client local data for knowledge fusion.

**MHeteroFL with Model Split.** These methods split models into feature extractors and predictors. Some [7, 10, 34, 36] share homogeneous feature extractors across clients and personalize predictors, while others (`LG-FedAvg` [27] and [20, 29]) do the opposite. Such methods expose part of the local model structures, which might not be acceptable if the models are proprietary IPs of the clients.

**MHeteroFL with Mutual Learning.** These methods (`FedAPEN` [37], `FML` [41], `FedKD` [45] and others [31, 22]) add a shared global homogeneous small model on top of each client's heterogeneous local model. For each local data sample, the distance of the outputs from these two models is used as the mutual loss to update model parameters. Nevertheless, the mutual loss only transfers limited knowledge between the two models, resulting in model performance bottlenecks.

The proposed `FedMRL` approach further optimizes mutual learning-based MHeteroFL by enhancing the knowledge transfer between the server and client models. It achieves personalized adaptive representation fusion and multi-perspective representation learning, thereby facilitating more knowledge interaction across the two models and improving model performance.

## 3 The Proposed `FedMRL` Approach

`FedMRL` aims to tackle data, system, and model heterogeneity in supervised learning tasks, where a central FL server coordinates $N$ FL clients to train heterogeneous local models. The server maintains a global homogeneous small model $\mathcal{G}(\theta)$ shared by all clients. Figure 2 depicts its workflow [2]:

① In each communication round, $K$ clients participate in FL (*i.e.*, the client participant rate $C = K/N$). The global homogeneous small model $\mathcal{G}(\theta)$ is broadcast to them.

② Each client $k$ holds a heterogeneous local model $\mathcal{F}_k(\omega_k)$ ($\mathcal{F}_k(\cdot)$ is the heterogeneous model structure, and $\omega_k$ are personalized model parameters). Client $k$ simultaneously trains the heterogeneous local model and the global homogeneous small model on local non-IID data $D_k$ ($D_k$ follows the non-IID distribution $P_k$) via personalized Matryoshka Representations Learning with a personalized representation projector $\mathcal{P}_k(\varphi_k)$ in an end-to-end manner.

③ The updated homogeneous small models are uploaded to the server for aggregation to produce a new global model for knowledge fusion across heterogeneous clients.

The objective of `FedMRL` is to minimize the sum of the loss from the combined models ($\mathcal{W}_k(w_k) = (\mathcal{G}(\theta) \circ \mathcal{F}_k(\omega_k)|\mathcal{P}_k(\varphi_k))$) on all clients, *i.e.*,

$$\min_{\theta, \omega_{0,\ldots,N-1}} \sum_{k=0}^{N-1} \ell\left(\mathcal{W}_k\left(D_k; (\theta \circ \omega_k \mid \varphi_k)\right)\right). \tag{1}$$

These steps repeat until each client's model converges. After FL training, a client uses its local combined model without the global header for inference. [3]

---

[2]Algorithm 1 in Appendix A describes the `FedMRL` algorithm.

[3]Appendix C.3 provides experimental evidence for inference model selection.

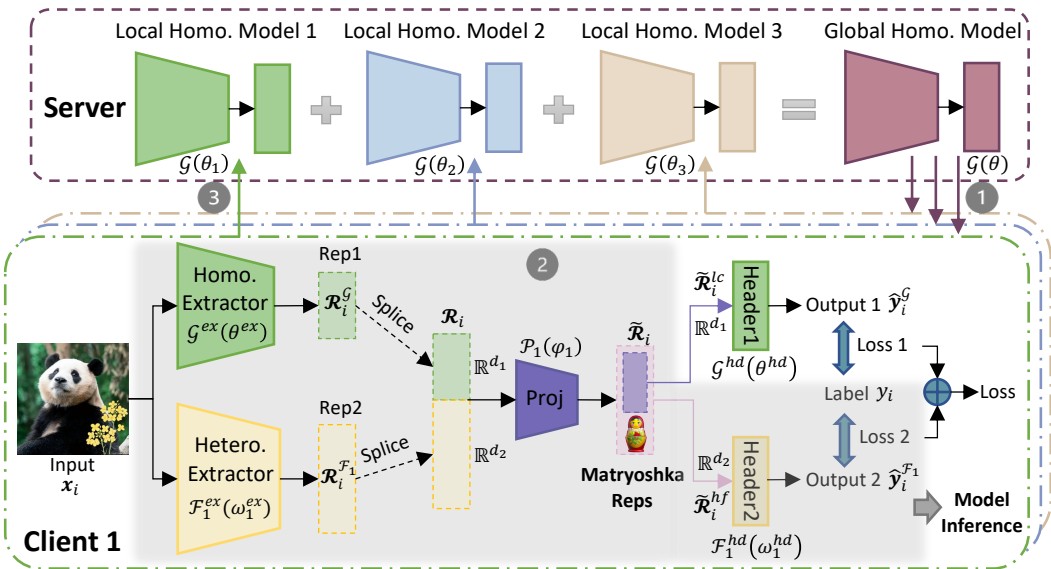

Figure 2: The workflow of FedMRL.

## 3.1 Adaptive Representation Fusion

We denote client $k$'s heterogeneous local model feature extractor as $\mathcal{F}_k^{ex}(\omega_k^{ex})$, and prediction header as $\mathcal{F}_k^{hd}(\omega_k^{hd})$. We denote the homogeneous global model feature extractor as $\mathcal{G}^{ex}(\theta^{ex})$ and prediction header as $\mathcal{G}^{hd}(\theta^{hd})$. Client $k$'s local personalized representation projector is denoted as $\mathcal{P}_k(\varphi_k)$. In the $t$-th communication round, client $k$ inputs its local data sample $(x_i, y_i) \in D_k$ into the two feature extractors to extract generalized and personalized representations as:

$$\mathcal{R}_i^{\mathcal{G}} = \mathcal{G}^{ex}(x_i; \theta^{ex,t-1}), \mathcal{R}_i^{\mathcal{F}_k} = \mathcal{F}_k^{ex}(x_i; \omega_k^{ex,t-1}). \tag{2}$$

The two extracted representations $\mathcal{R}_i^{\mathcal{G}} \in \mathbb{R}^{d_1}$ and $\mathcal{R}_i^{\mathcal{F}_k} \in \mathbb{R}^{d_2}$ are spliced as:

$$\mathcal{R}_i = \mathcal{R}_i^{\mathcal{G}} \circ \mathcal{R}_i^{\mathcal{F}_k}. \tag{3}$$

Then, the spliced representation is mapped into a fused representation by the lightweight representation projector $\mathcal{P}_k(\varphi_k^{t-1})$ as:

$$\widetilde{\mathcal{R}}_i = \mathcal{P}_k(\mathcal{R}_i; \varphi_k^{t-1}), \tag{4}$$

where the projector can be a one-layer linear model or multi-layer perceptron. The fused representation $\widetilde{\mathcal{R}}_i$ contains both generalized and personalized feature information. It has the same dimension as the client's local heterogeneous model representation $\mathbb{R}^{d_2}$, which ensures the representation dimension $\mathbb{R}^{d_2}$ and the client local heterogeneous model header parameter dimension $\mathbb{R}^{d_2 \times L}$ ($L$ is the label dimension) match.

The representation projector can be updated as the two models are being trained on local non-IID data. Hence, it achieves personalized representation fusion adaptive to local data distributions. Splicing the representations extracted by two feature extractors can keep the relative semantic space positions of the generalized and personalized representations, benefiting the construction of multi-granularity Matryoshka Representations. Owing to representation splicing, the representation dimensions of the two feature extractors can be different (*i.e.*, $d_1 \leq d_2$). Therefore, we can vary the representation dimension of the small homogeneous global model to improve the trade-off among model performance, storage requirement and communication costs.

In addition, each client's local model is treated as a black box by the FL server. When the server broadcasts the global homogeneous small model to the clients, each client can adjust the linear layer dimension of the representation projector to align it with the dimension of the spliced representation. In this way, different clients may hold different representation projectors. When a new model-agnostic client joins in FedMRL, it can adjust its representation projector structure for local model training. Therefore, FedMRL can accommodate FL clients owning local models with diverse structures.

## 3.2 Multi-Granular Representation Learning

To construct multi-dimensional and multi-granular Matryoshka Representations, we further extract a low-dimension coarse-granularity representation $\widetilde{\mathcal{R}}_i^{lc}$ and a high-dimension fine-granularity representation $\widetilde{\mathcal{R}}_i^{hf}$ from the fused representation $\widetilde{\mathcal{R}}_i$. They align with the representation dimensions $\{\mathbb{R}^{d_1}, \mathbb{R}^{d_2}\}$ of two feature extractors for matching the parameter dimensions $\{\mathbb{R}^{d_1 \times L}, \mathbb{R}^{d_2 \times L}\}$ of the two prediction headers,

$$\widetilde{\mathcal{R}}_i^{lc} = \widetilde{\mathcal{R}}_i^{1:d_1}, \widetilde{\mathcal{R}}_i^{hf} = \widetilde{\mathcal{R}}_i^{1:d_2}. \tag{5}$$

The embedded low-dimension coarse-granularity representation $\widetilde{\mathcal{R}}_i^{lc} \in \mathbb{R}^{d_1}$ incorporates coarse generalized and personalized feature information. It is learned by the global homogeneous model header $\mathcal{G}^{hd}(\theta^{hd,t-1})$ (parameter space: $\mathbb{R}^{d_1 \times L}$) with generalized prediction information to produce:

$$\hat{y}_i^{\mathcal{G}} = \mathcal{G}^{hd}(\widetilde{\mathcal{R}}_i^{lc}; \theta^{hd,t-1}). \tag{6}$$

The embedded high-dimension fine-granularity representation $\widetilde{\mathcal{R}}_i^{hf} \in \mathbb{R}^{d_2}$ carries finer generalized and personalized feature information, which is further processed by the heterogeneous local model header $\mathcal{F}_k^{hd}(\omega_k^{hd,t-1})$ (parameter space: $\mathbb{R}^{d_2 \times L}$) with personalized prediction information to generate:

$$\hat{y}_i^{\mathcal{F}_k} = \mathcal{F}_k^{hd}(\widetilde{\mathcal{R}}_i^{hf}; \omega_k^{hd,t-1}). \tag{7}$$

We compute the losses $\ell$ (*e.g.*, cross-entropy loss [63]) between the two outputs and the label $y_i$ as:

$$\ell_i^{\mathcal{G}} = \ell(\hat{y}_i^{\mathcal{G}}, y_i), \ \ell_i^{\mathcal{F}_k} = \ell(\hat{y}_i^{\mathcal{F}_k}, y_i). \tag{8}$$

Then, the losses of the two branches are weighted by their importance $m_i^{\mathcal{G}}$ and $m_i^{\mathcal{F}_k}$ and summed as:

$$\ell_i = m_i^{\mathcal{G}} \cdot \ell_i^{\mathcal{G}} + m_i^{\mathcal{F}_k} \cdot \ell_i^{\mathcal{F}_k}. \tag{9}$$

We set $m_i^{\mathcal{G}} = m_i^{\mathcal{F}_k} = 1$ by default to make the two models contribute equally to model performance. The complete loss $\ell_i$ is used to simultaneously update the homogeneous global small model, the heterogeneous client local model, and the representation projector via gradient descent:

$$\begin{aligned} \theta_k^t &\leftarrow \theta^{t-1} - \eta_\theta \nabla \ell_i, \\ \omega_k^t &\leftarrow \omega_k^{t-1} - \eta_\omega \nabla \ell_i, \\ \varphi_k^t &\leftarrow \varphi_k^{t-1} - \eta_\varphi \nabla \ell_i, \end{aligned} \tag{10}$$

where $\eta_\theta, \eta_\omega, \eta_\varphi$ are the learning rates of the homogeneous global small model, the heterogeneous local model and the representation projector. We set $\eta_\theta = \eta_\omega = \eta_\varphi$ by default to ensure stable model convergence. In this way, the generalized and personalized fused representation is learned from multiple perspectives, thereby improving model learning capability.

## 4 Convergence Analysis

Based on notations, assumptions and proofs in Appendix B, we analyse the convergence of FedMRL.

**Lemma 1** *Local Training. Given Assumptions 1 and 2, the loss of an arbitrary client's local model $w$ in local training round $(t+1)$ is bounded by:*

$$\mathbb{E}[\mathcal{L}_{(t+1)E}] \leq \mathcal{L}_{tE+0} + (\frac{L_1 \eta^2}{2} - \eta) \sum_{e=0}^{E} \|\nabla \mathcal{L}_{tE+e}\|_2^2 + \frac{L_1 E \eta^2 \sigma^2}{2}. \tag{11}$$

**Lemma 2** *Model Aggregation. Given Assumptions 2 and 3, after local training round $(t+1)$, a client's loss before and after receiving the updated global homogeneous small models is bounded by:*

$$\mathbb{E}[\mathcal{L}_{(t+1)E+0}] \leq \mathbb{E}[\mathcal{L}_{(t+1)E}] + \eta \delta^2. \tag{12}$$

**Theorem 1** *One Complete Round of FL. Given the above lemmas, for any client, after receiving the updated global homogeneous small model, we have:*

$$\mathbb{E}[\mathcal{L}_{(t+1)E+0}] \le \mathcal{L}_{tE+0} + (\frac{L_1\eta^2}{2} - \eta)\sum_{e=0}^{E}\|\nabla\mathcal{L}_{tE+e}\|_2^2 + \frac{L_1E\eta^2\sigma^2}{2} + \eta\delta^2. \qquad (13)$$

**Theorem 2** *Non-convex Convergence Rate of FedMRL. Given Theorem 1, for any client and an arbitrary constant $\epsilon > 0$, the following holds:*

$$\frac{1}{T}\sum_{t=0}^{T-1}\sum_{e=0}^{E-1}\|\nabla\mathcal{L}_{tE+e}\|_2^2 \le \frac{\frac{1}{T}\sum_{t=0}^{T-1}[\mathcal{L}_{tE+0} - \mathbb{E}[\mathcal{L}_{(t+1)E+0}]] + \frac{L_1E\eta^2\sigma^2}{2} + \eta\delta^2}{\eta - \frac{L_1\eta^2}{2}} < \epsilon,$$
$$s.t.\ \eta < \frac{2(\epsilon - \delta^2)}{L_1(\epsilon + E\sigma^2)}. \qquad (14)$$

Therefore, we conclude that any client's local model can converge at a non-convex rate of $\epsilon \sim \mathcal{O}(1/T)$ in `FedMRL` if the learning rates of the homogeneous small model, the client local heterogeneous model and the personalized representation projector satisfy the above conditions.

## 5   Experimental Evaluation

We implement `FedMRL` on Pytorch, and compare it with seven state-of-the-art MHeteroFL methods. The experiments are carried out over two benchmark supervised image classification datasets on 4 NVIDIA GeForce 3090 GPUs (24GB Memory).[4]

### 5.1   Experiment Setup

**Datasets.** The benchmark datasets adopted are CIFAR-10 and CIFAR-100 [5] [23], which are commonly used in FL image classification tasks for the evaluating existing MHeteroFL algorithms. CIFAR-10 has $60,000$ $32 \times 32$ colour images across 10 classes, with $50,000$ for training and $10,000$ for testing. CIFAR-100 has $60,000$ $32 \times 32$ colour images across 100 classes, with $50,000$ for training and $10,000$ for testing. We follow [40] and [37] to construct two types of non-IID datasets. Each client's non-IID data are further divided into a training set and a testing set with a ratio of $8 : 2$.

- **Non-IID (Class):** For CIFAR-10 with 10 classes, we randomly assign 2 classes to each FL client. For CIFAR-100 with 100 classes, we randomly assign 10 classes to each FL client. The fewer classes each client possesses, the higher the non-IIDness.

- **Non-IID (Dirichlet):** To produce more sophisticated non-IID data settings, for each class of CIFAR-10/CIFAR-100, we use a Dirichlet($\alpha$) function to adjust the ratio between the number of FL clients and the assigned data. A smaller $\alpha$ indicates more pronounced non-IIDness.

**Models.** We evaluate MHeteroFL algorithms under model-homogeneous and heterogeneous FL scenarios. `FedMRL`'s representation projector is a one-layer linear model (parameter space: $\mathbb{R}^{d2\times(d_1+d_2)}$).

- **Model-Homogeneous FL:** All clients train CNN-1 in Table 2 (Appendix C.1). The homogeneous global small models in `FML` and `FedKD` are also CNN-1. The extra homogeneous global small model in `FedMRL` is CNN-1 with a smaller representation dimension $d_1$ (*i.e.*, the penultimate linear layer dimension) than the CNN-1 model's representation dimension $d_2, d_1 \le d_2$.

- **Model-Heterogeneous FL:** The 5 heterogeneous models {CNN-1, ..., CNN-5} in Table 2 (Appendix C.1) are evenly distributed among FL clients. The homogeneous global small models in `FML` and `FedKD` are the smallest CNN-5 models. The homogeneous global small model in `FedMRL` is the smallest CNN-5 with a reduced representation dimension $d_1$ compared with the CNN-5 model representation dimension $d_2$, *i.e.*, $d_1 \le d_2$.

---

[4] `https://github.com/LipingYi/FedMRL`
[5] `https://www.cs.toronto.edu/%7Ekriz/cifar.html`

Table 1: Average test accuracy (%) in model-heterogeneous FL.

| FL Setting | N=10, C=100% | | N=50, C=20% | | N=100, C=10% | |
|---|---|---|---|---|---|---|
| Method | CIFAR-10 | CIFAR-100 | CIFAR-10 | CIFAR-100 | CIFAR-10 | CIFAR-100 |
| Standalone | 96.53 | 72.53 | 95.14 | 62.71 | 91.97 | 53.04 |
| LG-FedAvg [27] | 96.30 | 72.20 | 94.83 | 60.95 | 91.27 | 45.83 |
| FD [21] | 96.21 | - | - | - | - | - |
| FedProto [43] | 96.51 | 72.59 | 95.48 | 62.69 | 92.49 | 53.67 |
| FML [41] | 30.48 | 16.84 | - | 21.96 | - | 15.21 |
| FedKD [45] | 80.20 | 53.23 | 77.37 | 44.27 | 73.21 | 37.21 |
| FedAPEN [37] | - | - | - | - | - | - |
| FedMRL | **96.63** | **74.37** | **95.70** | **66.04** | **95.85** | **62.15** |
| FedMRL-*Best B.* | *0.10* | *1.78* | *0.22* | *3.33* | *3.36* | *8.48* |
| FedMRL-*Best S.C.B.* | *16.43* | *21.14* | *18.33* | *21.77* | *22.64* | *24.94* |

"-": failing to converge. "▣": the best MHeteroFL method. "▢ Best B.": the best baseline. "▢ Best S.C.B.": the best same-category (mutual learning-based MHeteroFL) baseline. The underscored values denote the largest accuracy improvement of FedMRL across 6 settings.

**Comparison Baselines.** We compare FedMRL with state-of-the-art algorithms belonging to the following three categories of MHeteroFL methods:

- Standalone. Each client trains its heterogeneous local model only with its local data.
- **Knowledge Distillation Without Public Data:** FD [21] and FedProto [43].
- **Model Split:** LG-FedAvg [27].
- **Mutual Learning:** FML [41], FedKD [45] and FedAPEN [37].

**Evaluation Metrics.** We evaluate MHeteroFL algorithms from the following three aspects:

- **Model Accuracy.** We record the test accuracy of each client's model in each round, and compute the average test accuracy.
- **Communication Cost.** We compute the number of parameters sent between the server and one client in one communication round, and record the required rounds for reaching the target average accuracy. The overall communication cost of one client for target average accuracy is the product between the cost per round and the number of rounds.
- **Computation Overhead.** We compute the computation FLOPs of one client in one communication round, and record the required communication rounds for reaching the target average accuracy. The overall computation overall for one client achieving the target average accuracy is the product between the FLOPs per round and the number of rounds.

**Training Strategy.** We search optimal FL hyperparameters and unique hyperparameters for all MHeteroFL algorithms. For FL hyperparameters, we test MHeteroFL algorithms with a $\{64, 128, 256, 512\}$ batch size, $\{1, 10\}$ epochs, $T = \{100, 500\}$ communication rounds and an SGD optimizer with a $0.01$ learning rate. The unique hyperparameter of FedMRL is the representation dimension $d_1$ of the homogeneous global small model, we vary $d_1 = \{100, 150, ..., 500\}$ to obtain the best-performing FedMRL.

## 5.2 Results and Discussion

We design three FL settings with different numbers of clients ($N$) and client participation rates ($C$): ($N = 10, C = 100\%$), ($N = 50, C = 20\%$), ($N = 100, C = 10\%$) for both model-homogeneous and model-heterogeneous FL scenarios.

### 5.2.1 Average Test Accuracy

Table 1 and Table 3 (Appendix C.2) show that FedMRL consistently outperforms all baselines under both model-heterogeneous or homogeneous settings. It achieves up to a $8.48\%$ improvement in average test accuracy compared with the best baseline under each setting. Furthermore, it achieves up to a $24.94\%$ average test accuracy improvement than the best same-category (*i.e.*, mutual learning-based MHeteroFL) baseline under each setting. These results demonstrate the superiority of FedMRL

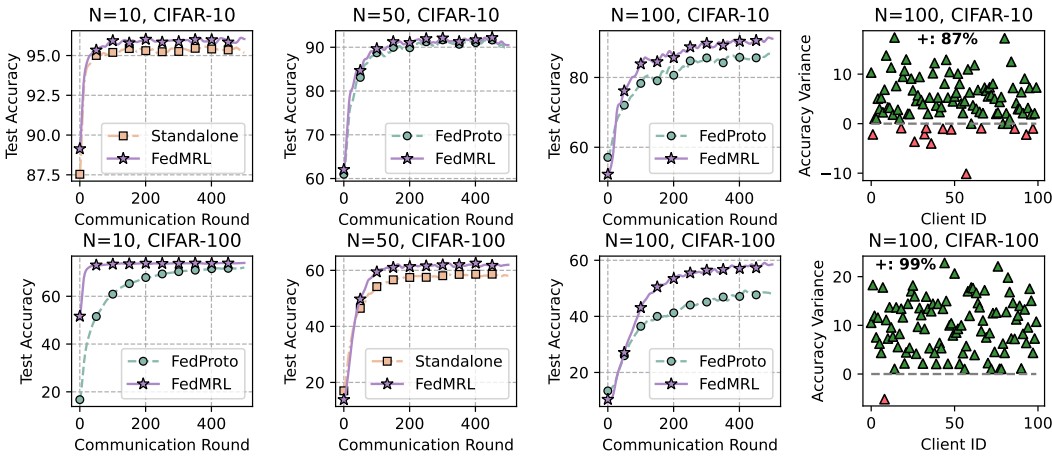

Figure 3: Left six: average test accuracy vs. communication rounds. Right two: individual clients' test accuracy (%) differences (`FedMRL` - `FedProto`).

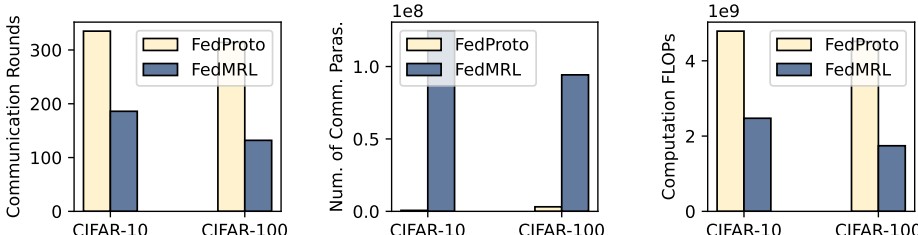

Figure 4: Communication rounds, number of communicated parameters, and computation FLOPs required to reach $90\%$ and $50\%$ average test accuracy targets on CIFAR-10 and CIFAR-100.

in model performance owing to its adaptive personalized representation fusion and multi-granularity representation learning capabilities. Figure 3(left six) shows that `FedMRL` consistently achieves faster convergence speed and higher average test accuracy than the best baseline under each setting.

### 5.2.2 Individual Client Test Accuracy

Figure 3(right two) shows the difference between the test accuracy achieved by `FedMRL` vs. the best-performing baseline `FedProto` (*i.e.*, `FedMRL` - `FedProto`) under ($N = 100, C = 10\%$) for each individual client. It can be observed that $87\%$ and $99\%$ of all clients achieve better performance under `FedMRL` than under `FedProto` on CIFAR-10 and CIFAR-100, respectively. This demonstrates that `FedMRL` possesses stronger personalization capability than `FedProto` owing to its adaptive personalized multi-granularity representation learning design.

### 5.2.3 Communication Cost

We record the communication rounds and the number of parameters sent per client to achieve $90\%$ and $50\%$ target test average accuracy on CIFAR-10 and CIFAR-100, respectively. Figure 4 (left) shows that `FedMRL` requires fewer rounds and achieves faster convergence than `FedProto`. Figure 4 (middle) shows that `FedMRL` incurs higher communication costs than `FedProto` as it transmits the full homogeneous small model, while `FedProto` only transmits each local seen-class average representation between the server and the client. Nevertheless, `FedMRL` with an optional smaller representation dimension ($d_1$) of the homogeneous small model still achieves higher communication efficiency than same-category mutual learning-based MHeteroFL baselines (`FML`, `FedKD`, `FedAPEN`) with a larger representation dimension.

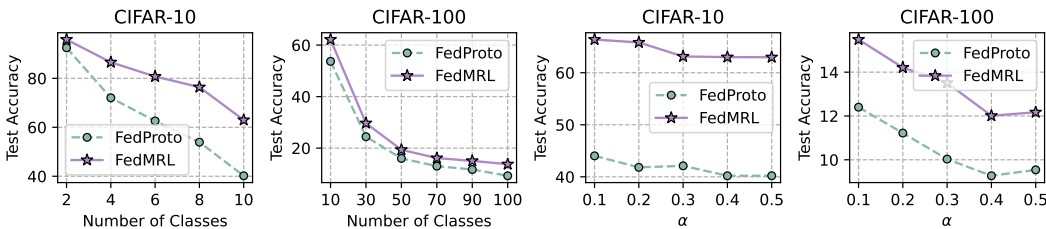

Figure 5: Robustness to non-IIDness (Class & Dirichlet).

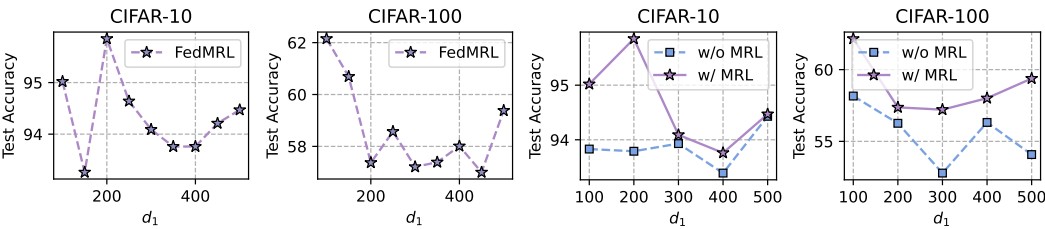

Figure 6: Left two: sensitivity analysis results. Right two: ablation study results.

#### 5.2.4 Computation Overhead

We also calculate the computation FLOPs consumed per client to reach $90\%$ and $50\%$ target average test accuracy on CIFAR-10 and CIFAR-100, respectively. Figure 4(right) shows that `FedMRL` incurs lower computation costs than `FedProto`, owing to its faster convergence (*i.e.*, fewer rounds) even with higher computation overhead per round due to the need to train an additional homogeneous small model and a linear representation projector.

### 5.3 Case Studies

#### 5.3.1 Robustness to Non-IIDness (Class)

We evaluate the robustness of `FedMRL` to different non-IIDnesses as a result of the number of classes assigned to each client under the $(N = 100, C = 10\%)$ setting. The fewer classes assigned to each client, the higher the non-IIDness. For CIFAR-10, we assign $\{2, 4, \ldots, 10\}$ classes out of total 10 classes to each client. For CIFAR-100, we assign $\{10, 30, \ldots, 100\}$ classes out of total 100 classes to each client. Figure 5(left two) shows that `FedMRL` consistently achieves higher average test accuracy than the best-performing baseline - `FedProto` on both datasets, demonstrating its robustness to non-IIDness by class.

#### 5.3.2 Robustness to Non-IIDness (Dirichlet)

We also test the robustness of `FedMRL` to various non-IIDnesses controlled by $\alpha$ in the Dirichlet function under the $(N = 100, C = 10\%)$ setting. A smaller $\alpha$ indicates a higher non-IIDness. For both datasets, we vary $\alpha$ in the range of $\{0.1, \ldots, 0.5\}$. Figure 5(right two) shows that `FedMRL` significantly outperforms `FedProto` under all non-IIDness settings, validating its robustness to Dirichlet non-IIDness.

#### 5.3.3 Sensitivity Analysis - $d_1$

`FedMRL` relies on a hyperparameter $d_1$ - the representation dimension of the homogeneous small model. To evaluate its sensitivity to $d_1$, we test `FedMRL` with $d_1 = \{100, 150, \ldots, 500\}$ under the $(N = 100, C = 10\%)$ setting. Figure 6(left two) shows that smaller $d_1$ values result in higher average test accuracy on both datasets. It is clear that a smaller $d_1$ also reduces communication and computation overheads, thereby helping `FedMRL` achieve the best trade-off among model performance, communication efficiency, and computational efficiency.

### 5.4 Ablation Study

We conduct ablation experiments to validate the usefulness of MRL. For `FedMRL` with MRL, the global header and the local header learn multi-granularity representations. For `FedMRL` without MRL, we directly input the representation fused by the representation projector into the client's local header for loss computation (*i.e.*, we do not extract Matryoshka Representations and remove the global header). Figure 6(right two) shows that `FedMRL` with MRL consistently outperforms `FedMRL` without MRL, demonstrating the effectiveness of the design to incorporate MRL into MHeteroFL. Besides, the accuracy gap between them decreases as $d_1$ rises. This shows that as the global and local headers learn increasingly overlapping representation information, the benefits of MRL are reduced.

## 6 Discussion

We discuss how `FedMRL` tackles heterogeneity and its privacy, communication and computation.

**Tackling Heterogeneity.** `FedMRL` allows each client to tailor its heterogeneous local model according to its system resources, which addresses system and model heterogeneity. Each client achieves multi-granularity representation learning adapting to local non-IID data distribution through a personalized heterogeneous representation projector, alleviating data heterogeneity.

**Privacy.** The server and clients only communicate the homogeneous small models. Since we do not limit the representation dimensions $d_1, d_2$ of the proxy homogeneous global model and the heterogeneous client model are the same, sharing the proxy homogeneous model does not disclose the representation dimension and structure of the heterogeneous client model. Meanwhile, local data are always stored by clients for local training, so local data privacy is also protected.

**Communication Cost.** The server and clients transmit homogeneous small models with fewer parameters than the client's heterogeneous local model, consuming significantly lower communication costs in one communication round compared with transmitting complete local models like `FedAvg`.

**Computational Overhead.** Besides training the heterogeneous local model, each client also trains the homogeneous global small model and a lightweight representation projector with far fewer parameters than the heterogeneous local model. The computational overhead in one round is slightly increased. Since we design personalized Matryoshka Representations learning adapting to local data distribution from multiple perspectives, the model learning capability is improved, accelerating model convergence and consuming fewer rounds. Therefore, the total computational cost is reduced.

## 7 Conclusion

This paper proposes a novel MHeteroFL approach - `FedMRL` - to jointly address data, system and model heterogeneity challenges in FL. The key design insight is the addition of a global homogeneous small model shared by FL clients for enhanced knowledge interaction among heterogeneous local models. Adaptive personalized representation fusion and multi-granularity Matryoshka Representations learning further boosts model learning capability. The client and the server only need to exchange the homogeneous small model, while the clients' heterogeneous local models and data remain unexposed, thereby enhancing the preservation of both model and data privacy. Theoretical analysis shows that `FedMRL` is guaranteed to converge over time. Extensive experiments demonstrate that `FedMRL` significantly outperforms state-of-the-art models regarding test accuracy, while incurring low communication and computation costs. [6]

## Acknowledgments and Disclosure of Funding

Xiaoguang Liu is supported by the National Science Foundation of China under Grant 62272252 & 62272253, and the Fundamental Research Funds for the Central Universities. Han Yu and Xiaoxiao Li are supported by the Ministry of Education, Singapore, under its Academic Research Fund Tier 1. This research is also supported, in part, by the RIE2025 Industry Alignment Fund – Industry Collaboration Projects (IAF-ICP) (Award I2301E0026), administered by A*STAR, as well as supported by Alibaba

---

[6]Appendix D elaborates `FedMRL`'s border impact and limitations.

Group and NTU Singapore through Alibaba-NTU Global e-Sustainability CorpLab (ANGEL); and National Research Foundation, Singapore and DSO National Laboratories under the AI Singapore Programme (AISG Award No: AISG2-RP-2020-019).

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

# A  Pseudo codes of `FedMRL`

---

**Algorithm 1:** `FedMRL`

---

**Input:** $N$, total number of clients; $K$, number of selected clients in one round; $T$, total number of rounds; $\eta_\omega$, learning rate of client local heterogeneous models; $\eta_\theta$, learning rate of homogeneous small model; $\eta_\varphi$, learning rate of the representation projector.

**Output:** client whole models removing the global header
$[\mathcal{G}(\theta^{ex,T-1}) \circ \mathcal{F}_0(\omega_0^{T-1}) | \mathcal{P}_0(\varphi_0^{T-1}), \ldots, \mathcal{G}(\theta^{ex,T-1}) \circ \mathcal{F}_{N-1}(\omega_{N-1}^{T-1}) | \mathcal{P}_{N-1}(\varphi_{N-1}^{T-1})].$

Randomly initialize the global homogeneous small model $\mathcal{G}(\theta^0)$, client local heterogeneous
  models $[\mathcal{F}_0(\omega_0^0), \ldots, \mathcal{F}_{N-1}(\omega_{N-1}^0)]$ and local heterogeneous representation projectors
  $[\mathcal{P}_0(\varphi_0^0), \ldots, \mathcal{P}_{N-1}(\varphi_{N-1}^0)].$

**for** *each round t=1,...,T-1* **do**

   |  // **Server Side**:
   |  $S^t \leftarrow$ Randomly sample $K$ clients from $N$ clients;
   |  Broadcast the global homogeneous small model $\theta^{t-1}$ to sampled $K$ clients;
   |  $\theta_k^t \leftarrow$ **ClientUpdate**$(\theta^{t-1})$;
   |  /* Aggregate Local Homogeneous Small Models */
   |  $\theta^t = \sum_{k=0}^{K-1} \frac{n_k}{n} \theta_k^t.$

   |  // **ClientUpdate**:
   |  Receive the global homogeneous small model $\theta^{t-1}$ from the server;
   |  **for** $k \in S^t$ **do**
   |    |  /* Local Training with MRL */
   |    |  **for** $(\boldsymbol{x}_i, y_i) \in D_k$ **do**
   |    |    |  $\mathcal{R}_i^{\mathcal{G}} = \mathcal{G}^{ex}(\boldsymbol{x}_i; \theta^{ex,t-1}), \mathcal{R}_i^{\mathcal{F}_k} = \mathcal{F}_k^{ex}(\boldsymbol{x}_i; \omega_k^{ex,t-1});$
   |    |    |  $\mathcal{R}_i = \mathcal{R}_i^{\mathcal{G}} \circ \mathcal{R}_i^{\mathcal{F}_k};$
   |    |    |  $\widetilde{\mathcal{R}}_i = \mathcal{P}_k(\mathcal{R}_i; \varphi_k^{t-1});$
   |    |    |  $\widetilde{\mathcal{R}}_i^{lc} = \widetilde{\mathcal{R}}_i^{1:d_1}, \widetilde{\mathcal{R}}_i^{hf} = \widetilde{\mathcal{R}}_i^{1:d_2};$
   |    |    |  $\hat{y}_i^{\mathcal{G}} = \mathcal{G}^{hd}(\widetilde{\mathcal{R}}_i^{lc}; \theta^{hd,t-1}); \hat{y}_i^{\mathcal{F}_k} = \mathcal{F}_k^{hd}(\omega_k^{hd,t-1});$
   |    |    |  $\ell_i^{\mathcal{G}} = \ell(\hat{y}_i^{\mathcal{G}}, y_i); \ell_i^{\mathcal{F}_k} = \ell(\hat{y}_i^{\mathcal{F}_k}, y_i);$
   |    |    |  $\ell_i = m_i^{\mathcal{G}} \cdot \ell_i^{\mathcal{G}} + m_i^{\mathcal{F}_k} \cdot \ell_i^{\mathcal{F}_k};$
   |    |    |  $\theta_k^t \leftarrow \theta^{t-1} - \eta_\theta \nabla \ell_i;$
   |    |    |  $\omega_k^t \leftarrow \omega_k^{t-1} - \eta_\omega \nabla \ell_i;$
   |    |    |  $\varphi_k^t \leftarrow \varphi_k^{t-1} - \eta_\varphi \nabla \ell_i;$
   |    |  **end**
   |    |  Upload updated local homogeneous small model $\theta_k^t$ to the server.
   |  **end**

**end**

---

# B  Theoretical Proofs

We first define the following additional notations. $t \in \{0, \ldots, T-1\}$ denotes the $t$-th round. $e \in \{0, 1, \ldots, E\}$ denotes the $e$-th iteration of local training. $tE + 0$ indicates that clients receive the global homogeneous small model $\mathcal{G}(\theta^t)$ from the server before the $(t+1)$-th round's local training. $tE + e$ denotes the $e$-th iteration of the $(t+1)$-th round's local training. $tE + E$ marks the ending of the $(t+1)$-th round's local training. After that, clients upload their updated local homogeneous small models to the server for aggregation. $\mathcal{W}_k(w_k)$ denotes the whole model trained on client $k$, including the global homogeneous small model $\mathcal{G}(\theta)$, the client $k$'s local heterogeneous model $\mathcal{F}_k(\omega_k)$, and the personalized representation projector $\mathcal{P}_k(\varphi_k)$. $\eta$ is the learning rate of the whole model trained on client $k$, including $\{\eta_\theta, \eta_\omega, \eta_\varphi\}$.

**Assumption 1** *Lipschitz Smoothness. The gradients of client $k$'s whole local model $w_k$ are L1–Lipschitz smooth [43],*

$$\|\nabla \mathcal{L}_k^{t_1}(w_k^{t_1}; \boldsymbol{x}, y) - \nabla \mathcal{L}_k^{t_2}(w_k^{t_2}; \boldsymbol{x}, y)\| \leq L_1 \|w_k^{t_1} - w_k^{t_2}\|,$$
$$\forall t_1, t_2 > 0, k \in \{0, 1, \ldots, N-1\}, (\boldsymbol{x}, y) \in D_k. \tag{15}$$

*The above formulation can be re-expressed as:*

$$\mathcal{L}_k^{t_1} - \mathcal{L}_k^{t_2} \leq \langle \nabla \mathcal{L}_k^{t_2}, (w_k^{t_1} - w_k^{t_2}) \rangle + \frac{L_1}{2} \|w_k^{t_1} - w_k^{t_2}\|_2^2. \tag{16}$$

**Assumption 2** *Unbiased Gradient and Bounded Variance. Client $k$'s random gradient $g_{w,k}^t = \nabla \mathcal{L}_k^t(w_k^t; \mathcal{B}_k^t)$ ($\mathcal{B}$ is a batch of local data) is unbiased,*

$$\mathbb{E}_{\mathcal{B}_k^t \subseteq D_k}[g_{w,k}^t] = \nabla \mathcal{L}_k^t(w_k^t), \tag{17}$$

*and the variance of random gradient $g_{w,k}^t$ is bounded by:*

$$\mathbb{E}_{\mathcal{B}_k^t \subseteq D_k}[\|\nabla \mathcal{L}_k^t(w_k^t; \mathcal{B}_k^t) - \nabla \mathcal{L}_k^t(w_k^t)\|_2^2] \leq \sigma^2. \tag{18}$$

**Assumption 3** *Bounded Parameter Variation. The parameter variations of the homogeneous small model $\theta_k^t$ and $\theta^t$ before and after aggregation at the FL server are bounded by:*

$$\|\theta^t - \theta_k^t\|_2^2 \leq \delta^2. \tag{19}$$

## B.1 Proof of Lemma 1

**Proof 1** *An arbitrary client $k$'s local whole model $w$ can be updated by $w_{t+1} = w_t - \eta g_{w,t}$ in the $(t+1)$-th round, and following Assumption 1, we can obtain*

$$\mathcal{L}_{tE+1} \leq \mathcal{L}_{tE+0} + \langle \nabla \mathcal{L}_{tE+0}, (w_{tE+1} - w_{tE+0}) \rangle + \frac{L_1}{2} \|w_{tE+1} - w_{tE+0}\|_2^2$$
$$= \mathcal{L}_{tE+0} - \eta \langle \nabla \mathcal{L}_{tE+0}, g_{w,tE+0} \rangle + \frac{L_1 \eta^2}{2} \|g_{w,tE+0}\|_2^2. \tag{20}$$

*Taking the expectation of both sides of the inequality concerning the random variable $\xi_{tE+0}$,*

$$\mathbb{E}[\mathcal{L}_{tE+1}] \leq \mathcal{L}_{tE+0} - \eta \mathbb{E}[\langle \nabla \mathcal{L}_{tE+0}, g_{w,tE+0} \rangle] + \frac{L_1 \eta^2}{2} \mathbb{E}[\|g_{w,tE+0}\|_2^2]$$
$$\overset{(a)}{=} \mathcal{L}_{tE+0} - \eta \|\nabla \mathcal{L}_{tE+0}\|_2^2 + \frac{L_1 \eta^2}{2} \mathbb{E}[\|g_{w,tE+0}\|_2^2]$$
$$\overset{(b)}{\leq} \mathcal{L}_{tE+0} - \eta \|\nabla \mathcal{L}_{tE+0}\|_2^2 + \frac{L_1 \eta^2}{2} (\mathbb{E}[\|g_{w,tE+0}\|]_2^2 + \mathrm{Var}(g_{w,tE+0}))$$
$$\overset{(c)}{=} \mathcal{L}_{tE+0} - \eta \|\nabla \mathcal{L}_{tE+0}\|_2^2 + \frac{L_1 \eta^2}{2} (\|\nabla \mathcal{L}_{tE+0}\|_2^2 + \mathrm{Var}(g_{w,tE+0}))$$
$$\overset{(d)}{\leq} \mathcal{L}_{tE+0} - \eta \|\nabla \mathcal{L}_{tE+0}\|_2^2 + \frac{L_1 \eta^2}{2} (\|\nabla \mathcal{L}_{tE+0}\|_2^2 + \sigma^2)$$
$$= \mathcal{L}_{tE+0} + (\frac{L_1 \eta^2}{2} - \eta) \|\nabla \mathcal{L}_{tE+0}\|_2^2 + \frac{L_1 \eta^2 \sigma^2}{2}. \tag{21}$$

*(a), (c), (d) follow Assumption 2 and (b) follows $Var(x) = \mathbb{E}[x^2] - (\mathbb{E}[x])^2$.*

*Taking the expectation of both sides of the inequality for the model $w$ over $E$ iterations, we obtain*

$$\mathbb{E}[\mathcal{L}_{tE+1}] \leq \mathcal{L}_{tE+0} + (\frac{L_1 \eta^2}{2} - \eta) \sum_{e=1}^{E} \|\nabla \mathcal{L}_{tE+e}\|_2^2 + \frac{L_1 E \eta^2 \sigma^2}{2}. \tag{22}$$

## B.2 Proof of Lemma 2

**Proof 2**

$$\mathcal{L}_{(t+1)E+0} = \mathcal{L}_{(t+1)E} + \mathcal{L}_{(t+1)E+0} - \mathcal{L}_{(t+1)E}$$
$$\overset{(a)}{\approx} \mathcal{L}_{(t+1)E} + \eta \|\theta_{(t+1)E+0} - \theta_{(t+1)E}\|_2^2 \tag{23}$$
$$\overset{(b)}{\leq} \mathcal{L}_{(t+1)E} + \eta \delta^2.$$

*(a): we can use the gradient of parameter variations to approximate the loss variations,* i.e., $\Delta\mathcal{L} \approx \eta \cdot \|\Delta\theta\|_2^2$. *(b) follows Assumption 3.*

*Taking the expectation of both sides of the inequality to the random variable $\xi$, we obtain*

$$\mathbb{E}[\mathcal{L}_{(t+1)E+0}] \leq \mathbb{E}[\mathcal{L}_{(t+1)E}] + \eta\delta^2. \tag{24}$$

## B.3 Proof of Theorem 1

**Proof 3** *Substituting Lemma 1 into the right side of Lemma 2's inequality, we obtain*

$$\mathbb{E}[\mathcal{L}_{(t+1)E+0}] \leq \mathcal{L}_{tE+0} + (\frac{L_1\eta^2}{2} - \eta)\sum_{e=0}^{E}\|\nabla\mathcal{L}_{tE+e}\|_2^2 + \frac{L_1E\eta^2\sigma^2}{2} + \eta\delta^2. \tag{25}$$

## B.4 Proof of Theorem 2

**Proof 4** *Interchanging the left and right sides of Eq. (25), we obtain*

$$\sum_{e=0}^{E}\|\nabla\mathcal{L}_{tE+e}\|_2^2 \leq \frac{\mathcal{L}_{tE+0} - \mathbb{E}[\mathcal{L}_{(t+1)E+0}] + \frac{L_1E\eta^2\sigma^2}{2} + \eta\delta^2}{\eta - \frac{L_1\eta^2}{2}}. \tag{26}$$

*Taking the expectation of both sides of the inequality over rounds $t = [0, T-1]$ to $w$, we obtain*

$$\frac{1}{T}\sum_{t=0}^{T-1}\sum_{e=0}^{E-1}\|\nabla\mathcal{L}_{tE+e}\|_2^2 \leq \frac{\frac{1}{T}\sum_{t=0}^{T-1}[\mathcal{L}_{tE+0} - \mathbb{E}[\mathcal{L}_{(t+1)E+0}]] + \frac{L_1E\eta^2\sigma^2}{2} + \eta\delta^2}{\eta - \frac{L_1\eta^2}{2}}. \tag{27}$$

*Let $\Delta = \mathcal{L}_{t=0} - \mathcal{L}^* > 0$, then $\sum_{t=0}^{T-1}[\mathcal{L}_{tE+0} - \mathbb{E}[\mathcal{L}_{(t+1)E+0}]] \leq \Delta$, we can get*

$$\frac{1}{T}\sum_{t=0}^{T-1}\sum_{e=0}^{E-1}\|\nabla\mathcal{L}_{tE+e}\|_2^2 \leq \frac{\frac{\Delta}{T} + \frac{L_1E\eta^2\sigma^2}{2} + \eta\delta^2}{\eta - \frac{L_1\eta^2}{2}}. \tag{28}$$

*If the above equation converges to a constant $\epsilon$,* i.e.,

$$\frac{\frac{\Delta}{T} + \frac{L_1E\eta^2\sigma^2}{2} + \eta\delta^2}{\eta - \frac{L_1\eta^2}{2}} < \epsilon, \tag{29}$$

*then*

$$T > \frac{\Delta}{\epsilon(\eta - \frac{L_1\eta^2}{2}) - \frac{L_1E\eta^2\sigma^2}{2} - \eta\delta^2}. \tag{30}$$

*Since $T > 0, \Delta > 0$, we can get*

$$\epsilon(\eta - \frac{L_1\eta^2}{2}) - \frac{L_1E\eta^2\sigma^2}{2} - \eta\delta^2 > 0. \tag{31}$$

*Solving the above inequality yields*

$$\eta < \frac{2(\epsilon - \delta^2)}{L_1(\epsilon + E\sigma^2)}. \tag{32}$$

*For $\epsilon - \delta^2$, Assumption 3 assumes that the parameter variations of the homogeneous small model $\theta_k^t$ and $\theta^t$ before and after aggregation are bounded by $|\theta^t - \theta_k^t|_2^2 \leq \delta^2$. $\theta_k^t = \theta^{t-1} - \eta\sum_{e=0}^{E-1}g_{\theta^{t-1}}$, so $|\theta^t - \theta_k^t|_2^2 = |\theta^t - \theta^{t-1} + \eta\sum_{e=0}^{E-1}g_{\theta^{t-1}}|_2^2 \approx \eta^2\sum_{e=0}^{E-1}|g_{\theta^{t-1}}|_2^2$, considering that the global homogeneous small models during two consecutive rounds have relatively small variations compared with parameter variations between the local and global homogeneous model. Eq. (28) and (29) define $\epsilon$ as the upper bound of the average gradient of the local training whole model (including homogeneous small model, heterogeneous client model and the local representation projector)*

*during $T$ rounds and $E$ epochs per round, i.e., $\frac{1}{T}\sum_{t=0}^{T-1}\sum_{e=0}^{E-1}|\mathcal{L}_{tE+e}|_2^2 < \epsilon$, we can simplify it to $\sum_{e=0}^{E-1}|\mathcal{L}_{tE+e}|_2^2 < \epsilon$. Since the homogeneous model $\theta$ is only one part of the local training whole model, so $\epsilon > \sum_{e=0}^{E-1}|\mathcal{L}_{tE+e}|_2^2 > \sum_{e=0}^{E-1}|g_{\theta^{t-1}}|_2^2$. Since we use the leaning rate $\eta \in (0,1)$, $\eta^2 \in (0,1)$, so $\epsilon > \sum_{e=0}^{E-1}|\mathcal{L}_{tE+e}|_2^2 > \sum_{e=0}^{E-1}|g_{\theta^{t-1}}|_2^2 > \eta^2\sum_{e=0}^{E-1}|g_{\theta^{t-1}}|_2^2$. Since $\delta^2$ is the upper bound of $\eta^2\sum_{e=0}^{E-1}|g_\theta^{t-1}|_2^2$, so $\epsilon > \delta^2$ and $\epsilon - \delta^2 > 0$.*

*Since $L_1$, $\epsilon$, $\sigma^2$, $\epsilon - \delta^2$ are all constants greater than 0, $\eta$ has solutions. Therefore, when the learning rate $\eta = \{\eta_\theta, \eta_\omega, \eta_\varphi\}$ satisfies the above condition, any client's local whole model can converge. Since all terms on the right side of Eq. (28) except for $1/T$ are constants, hence `FedMRL`'s non-convex convergence rate is $\epsilon \sim \mathcal{O}(1/T)$.*

## C    More Experimental Details

Here, we provide more experimental details of used model structures, more experimental results of model-homogeneous FL scenarios, and also the experimental evidence of inference model selection.

### C.1    Model Structures

Table 2 shows the structures of models used in experiments.

Table 2: Structures of 5 heterogeneous CNN models.

| Layer Name | CNN-1 | CNN-2 | CNN-3 | CNN-4 | CNN-5 |
|---|---|---|---|---|---|
| Conv1 | 5×5, 16 | 5×5, 16 | 5×5, 16 | 5×5, 16 | 5×5, 16 |
| Maxpool1 | 2×2 | 2×2 | 2×2 | 2×2 | 2×2 |
| Conv2 | 5×5, 32 | 5×5, 16 | 5×5, 32 | 5×5, 32 | 5×5, 32 |
| Maxpool2 | 2×2 | 2×2 | 2×2 | 2×2 | 2×2 |
| FC1 | 2000 | 2000 | 1000 | 800 | 500 |
| FC2 | 500 | 500 | 500 | 500 | 500 |
| FC3 | 10/100 | 10/100 | 10/100 | 10/100 | 10/100 |
| model size | 10.00 MB | 6.92 MB | 5.04 MB | 3.81 MB | 2.55 MB |

Note: $5 \times 5$ denotes kernel size. 16 or 32 are filters in convolutional layers.

### C.2    Homogeneous FL Results

Table 3 presents the results of `FedMRL` and baselines in model-homogeneous FL scenarios.

Table 3: Average test accuracy (%) in model-homogeneous FL.

| FL Setting | N=10, C=100% | | N=50, C=20% | | N=100, C=10% | |
|---|---|---|---|---|---|---|
| Method | CIFAR-10 | CIFAR-100 | CIFAR-10 | CIFAR-100 | CIFAR-10 | CIFAR-100 |
| Standalone | 96.35 | 74.32 | 95.25 | 62.38 | 92.58 | 54.93 |
| LG-FedAvg [27] | 96.47 | 73.43 | 94.20 | 61.77 | 90.25 | 46.64 |
| FD [21] | 96.30 | - | - | - | - | - |
| FedProto [43] | 95.83 | 72.79 | 95.10 | 62.55 | 91.19 | 54.01 |
| FML [41] | 94.83 | 70.02 | 93.18 | 57.56 | 87.93 | 46.20 |
| FedKD [45] | 94.77 | 70.04 | 92.93 | 57.56 | 90.23 | 50.99 |
| FedAPEN [37] | 95.38 | 71.48 | 93.31 | 57.62 | 87.97 | 46.85 |
| FedMRL | **96.71** | **74.52** | **95.76** | **66.46** | **95.52** | **60.64** |
| FedMRL-*Best B.* | *0.24* | *0.20* | *0.51* | *3.91* | *2.94* | *5.71* |
| FedMRL-*Best S.C.B.* | *1.33* | *3.04* | *2.45* | *8.84* | *5.29* | *9.65* |

"-": failing to converge. "▨": the best MHeteroFL method. "▢ Best B.": the best baseline. "▢ Best S.C.B.": the best same-category (mutual learning-based MHeteroFL) baseline. The underscored values denote the largest accuracy improvement of `FedMRL` across 6 settings.

### C.3    Inference Model Comparison

There are 4 alternative models for model inference in `FedMRL`: (1) mix-small (the combination of the homogeneous small model, the client heterogeneous model's feature extractor, and the representation projector, *i.e.*, removing the local header), (2) mix-large (the combination of the homogeneous small

model's feature extractor, the client heterogeneous model, and the representation projector, *i.e.*, removing the global header), (3) single-small (the homogeneous small model), (4) single-large (the client heterogeneous model). We compare their model performances under $(N = 100, C = 10\%)$ settings. Figure 7 presents that mix-small has a similar accuracy to mix-large which is used as the default inference model, and they significantly outperform the single homogeneous small model and the single heterogeneous client model. Therefore, users can choose mix-small or mix-large for model inference based on their inference costs in practical applications.

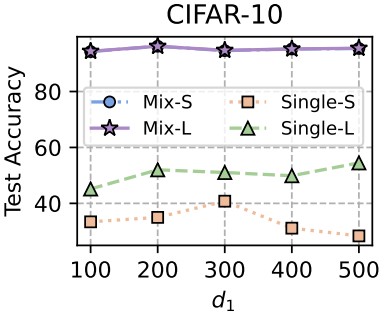 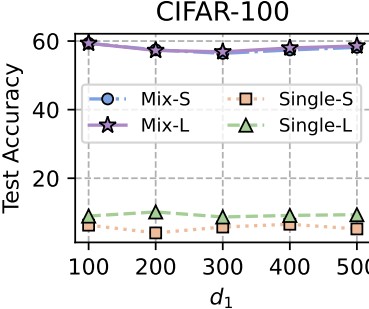

Figure 7: Accuracy of four optional inference models: mix-small (the whole model without the local header), mix-large (the whole model without the global header), single-small (the homogeneous small model), single-large (the client heterogeneous model).

## D  Broader Impacts and Limitations

**Broader Impacts.** FedMRL improves model performance, communication and computational efficiency for heterogeneous federated learning while effectively protecting the privacy of the client heterogeneous local model and non-IID data. It can be applied in various practical FL applications.

**Limitations.** The multi-granularity embedded representations within Matryoshka Representations are processed by the global small model's header and the local client model's header, respectively. This increases the storage cost, communication costs and training overhead for the global header even though it only involves one linear layer. In future work, we will follow the more effective Matryoshka Representation learning method (MRL-E) [24], removing the global header and only using the local model header to process multi-granularity Matryoshka Representations simultaneously, to enable a better trade-off among model performance and costs of storage, communication and computation.

