# OpenReview forum: "Federated Model Heterogeneous Matryoshka Representation Learning"
_NeurIPS.cc/2024/Conference — NeurIPS 2024 poster_

### Official Review · Reviewer_b3UH · 2024-06-28

**Soundness:** 2
**Presentation:** 3
**Contribution:** 3
**Rating:** 5
**Confidence:** 4

**Summary:**

This paper introduces FedMRL, a method based on distillation to mitigate the model heterogeneity issue in Federated Learning (FL). FedMRL operates by learning a small proxy homogeneous global model in a federated manner and distilling knowledge from it to heterogeneous client models. To enhance representation knowledge interaction between the homogeneous global model and the heterogeneous client local model, the authors employ a Matryoshka Representation Learning (MRL) approach, generating multi-dimensional and multi-granular representations. Theoretical analysis and experiments demonstrate the effectiveness of FedMRL.

**Strengths:**

1. The fusion of representations from the global and local models into a single representation vector, followed by their detachment in a Matryoshka manner, is intriguing and inspiring.
2. The writing is good and easy to follow.
3. Transmitting a global model with a relatively lower feature dimension is promising and can reduce communication overhead compared to using a similar dimension as local models.

**Weaknesses:**

1. The authors highlight the limitations of leveraging the training loss between server and client models (incurring high communication and computation costs) and reference the papers FedKD and FML in lines 39-41. However, the proposed FedMRL also falls within this category by sharing a proxy global model. Refer to Figure 4 for an illustration of FedMRL's high communication costs in the MHeteroFL domain, which also conflicts with the statement "low communication costs" in line 68. Additionally, there is a lack of numerical results regarding communication and computation costs between FedMRL and similar methods (FedKD and FML).
2. Two datasets for only image tasks are insufficient in FL.
3. The client models utilized in the experiments are not sufficiently heterogeneous, as they consist of CNN networks with identical numbers of Conv and FC layers. The variations are limited to the channels in Conv2 and the neural count in FC1. This setup lacks the persuasiveness needed to demonstrate FedMRL's effectiveness in MHeteroFL, especially considering that model architectures can significantly differ in size and structure, as noted in [1]. Can FedMRL accommodate settings involving CNNs and Vision Transformers (ViTs) on clients? Additionally, the considered CNNs are overly simple and small for a comprehensive evaluation.
4. There is only one baseline for the model split category, and it's worth noting that FedGH[2] also falls within this category.
5. The details of computing FLOPs are missing.
6. In the "Proof of Theorem 2" section, obtaining Eq. (31) directly from Eq. (30) is not feasible, as the right side is $\frac{\Delta}{T}$, not 0 as suggested by Eq. (30). Additionally, the existence of solutions for $\eta$ in Eq. (32) may be compromised if $\epsilon < \delta^2$, contradicting the conclusion in line 458 and potentially undermining the convergence guarantee.
7. The privacy analysis presented in lines 486-490 lacks sufficient substantiation. It would benefit from either theoretical analysis or experimental results. Without further analysis, it's challenging to accept the claim that "representation splicing enables the structures of the homogeneous global model and the heterogeneous local model to be not related," especially considering that the global and local models are trained together with a shared representation projector.

[1] Zhang, Jianqing, et al. "Fedtgp: Trainable global prototypes with adaptive-margin-enhanced contrastive learning for data and model heterogeneity in federated learning." Proceedings of the AAAI Conference on Artificial Intelligence. Vol. 38. No. 15. 2024.

[2] Yi, Liping, et al. "FedGH: Heterogeneous federated learning with generalized global header." Proceedings of the 31st ACM International Conference on Multimedia. 2023.

**Questions:**

See weaknesses.

**Limitations:**

The authors adequately addressed the limitations.

---

> ### Author Rebuttal · Authors · 2024-08-06
>
> **W1:**
> - **W1.a:** Apologize for the confusion. Our comment on the limitations of **existing strategies**, e.g., FedKD and FML, where they communicate the homogeneous proxy model. Although FedMRL falls in this category, it improved communication and computation efficiency by reducing the representation dimension of the homogeneous proxy model without sacrificing the performance through multi-dimension Matryoshka representation learning.
>
> - **W1.b:** For conciseness, we only compared with the best-performed baseline FedProto wrt averaged accuracy in Sec 5.2.
> First, we want to clarify that, although FedProto achieved lower communication costs, it requires more communication rounds and computational FLOPS in Figure 4. Also, it underperformed FedMRL by 3.36% and 8.48% on CIFAR-10 and CIFAR-100 (Table 1).
> Second, we wish to highlight that compared with the other baselines, FedMRL achieves relatively lower communication costs. As shown in **Table X2**, FML, FedKD, and FedAPEN belonging to the same category as FedMRL either do not converge or consume both higher communication and computation costs and also perform lower accuracy than FedMRL.
>
> - **W1.c:** We did not include the comparison with FML and FedKD as we only considered the methods that reach 90% and 50% target accuracy on CIFAR-10 and CIFAR-100 in lines 253-254, but they did not reach the targeted accuracies. We provide extra numerical results when reaching lower 70% and 30% target accuracy on CIFAR-10 and CIFAR-100 in **Table X3**. FedMRL still performs significantly lower communication and computational overheads than these same-category methods.
>
> **W2:** We selected two standard benchmark datasets to demonstrate the effectiveness of our methods following the existing literature [1,2] for fair comparison. We are grateful that our efforts on extensive and abundant experiments have been recognized by Reviewer 6thni and DqWb.
>
> To address your concerns and demonstrate the generalizability of our FedMRL to different datasets, we also add experiments for a next-word prediction (NLP) task with a large real-world non-IID dataset - Stack Overflow and heterogeneous LSTM models across 100 clients. As shown in **Table X4**, our method improves over the baseline methods by a significant margin.
>
> **W3:** We followed [1,2] to design the model heterogenities. And Yes,  FedMRL allows an arbitrary heterogeneous client model structure.
>
> To demonstrate the versatility of FedMRL,  we add experiments on CIFAR-100 with 100 clients including ResNet-{4,6,8,10,18,34,50,101,152} following FedTGP, CNN, and ViT. **Table X1** again shows the best model performance of FedMRL.
>
> **W4:** FedGH needs clients to upload labels to the server for training the shared global prediction header, which may be prohibited in some label-privacy-sensitive FL scenarios [3], so we did not compare it in the submission version. Nevertheless, we have discussed it in the related work section.
>
> We add experiments for FedGH in **Table X5**, which shows that FedMRL still maintains the best model performance.
>
> **W5:** We calculate the average FLOPs of heterogeneous CNN models consumed by forward inference and backwards updating in one iteration of local training, following the traditional FLOPs calculation rules of convolutional and linear layers. We then record the communication rounds required to reach the specified target accuracy and use the product between them as the total computational FLOPs.
>
> **W6:** There are some misunderstandings. The two steps of theoretical derivations definitely hold. Here, we give a detailed analysis.
> - As we stated $T>0$ and $\Delta>0$, then $\frac{\Delta}{T}>0$, so the derivation from Eq. (30) to Eq. (31) holds.
> - For $\epsilon>\delta^2$, we supplement additional analysis. Assumption 3 assumes that the parameter variations of the homogeneous small model $\theta_k^t$ and $\theta^t$ before and after aggregation are bounded by $|\theta^t-\theta_k^t| _ 2^2 \leq \delta^2$. $\theta_k^t=\theta^{t-1}-\eta\sum_{e=0}^{E-1}g_{\theta^{t-1}}$, so $|\theta^t-\theta_k^t| _ 2^2=|\theta^t-\theta^{t-1}+\eta \sum_{e=0}^{E-1} g_{\theta^{t-1}}| _ 2^2 \approx \eta^2 \sum_{e=0}^{E-1}|g_{\theta^{t-1}}| _ 2^2$, considering that the global homogeneous small models during two consecutive rounds have relatively small variations compared with parameter variations between the local and global homogeneous model.
> Eq. (28) and (29) define $\epsilon$ as the upper bound of the average gradient of the local training whole model (including homogeneous small model, heterogeneous client model and the local representation projector) during $T$ rounds and $E$ epochs per round, i.e.,$\frac{1}{T} \sum_{t=0}^{T-1} \sum_{e=0}^{E-1}|\mathcal{L} _ {t E+e}| _ 2^2<\epsilon$, we can simplify it to $\sum_{e=0}^{E-1}|\mathcal{L} _ {t E+e}| _ 2^2<\epsilon$. Since the homogeneous model $\theta$ is only one part of the local training whole model, so $\epsilon>\sum_{e=0}^{E-1}|\mathcal{L} _ {t E+e}| _ 2^2>\sum_{e=0}^{E-1}|g_{\theta^{t-1}}| _ 2^2$. Since we use the leaning rate $\eta\in(0,1)$, $\eta^2\in(0,1)$, so $\epsilon>\sum_{e=0}^{E-1}|\mathcal{L} _ {t E+e}| _ 2^2>\sum_{e=0}^{E-1}|g_{\theta^{t-1}}| _ 2^2>\eta^2 \sum_{e=0}^{E-1}|g_{\theta^{t-1}}| _ 2^2$. Since $\delta^2$ is the upper bound of $\eta^2 \sum_{e=0}^{E-1}|g _ \theta^{t-1}| _ 2^2$, so $\epsilon>\delta^2$.
>
>
> **W7:** Sorry for this vague description. To make a clear understanding, we re-write this sentence to “we do not limit the representation dimensions $d_1, d_2$ of the proxy homogeneous global model and the heterogeneous client model are the same, so sharing the proxy homogeneous model does not disclose the representation dimension and structure of the heterogeneous client model. ”
>
>
> [1] FedGH: Heterogeneous Federated Learning with Generalized Global Header.
>
> [2] FedAPEN: Personalized Cross-silo Federated Learning with Adaptability to Statistical Heterogeneity.
>
> [3] One-Shot Federated Learning with Label Differential Privacy.

---

> ### Comment · Reviewer_b3UH · 2024-08-10
> **Reply to authors**
>
> Thank you for your detailed responses, especially for the additional analysis to prove $\epsilon > \delta^2$. I have raised the score to the positive side.

---

> > ### Author Response · Authors · 2024-08-10
> >
> > Thank you very much for your supportive feedback on our response. We indeed highly appreciate your in-depth thought and your valuable time. Always happy to duscuss if additional clarification is required.

---

### Official Review · Reviewer_NjYm · 2024-07-05

**Soundness:** 3
**Presentation:** 3
**Contribution:** 2
**Rating:** 6
**Confidence:** 4

**Summary:**

The authors study the model heterogeneity challenge in federated learning using Matryoshka representation learning. It requires that the global model and the local models share one common part inspired by the two key modules: adaptive representation fusion and multi-granularity representation learning. They provide the experiment results and derive the non-convex convergence rate of the algorithm.

**Strengths:**

1, The model heterogeneity is one of the emerging challenges in the federated learning domain. The proposed approach avoids releasing the local model directly and solve the heterogeneous model cooperation issue.

2, This work covers comprehensive related works and the presentation is easy to follow.

3, They provide theoretical analysis in the paper and appendix.

**Weaknesses:**

1. The idea of sharing common parts of models in FL was introduced in FedGH[1], which all the clients and the server share an identical header. The idea of exchanging the small shared model was introduced in ProxyFL[2]. The authors are suggested to consider discussing these two works and emphasize the main novelty of contributions of the proposed approach.

2, This approach increase extra computational cost at the client side. The authors are suggested to provide some qualitative results and some quantitive analysis.

3, The CNN model structures are hand-crafted. The reviewer is wondering if the approach can work with other common-used CNN models and how they would perform.


[1] FedGH: Heterogeneous federated learning with generalized global header

[2] Decentralized federated learning through proxy model sharing

**Questions:**

1, Could you please address the concerns at the weakness part?

2, Could you please emphasize the motivation that why you combine Matryoshka representation learning with federated learning?

3, At the local client, is the optimization conducted in a end-to-end way or step-by-step way?

**Limitations:**

Multiple-run experiment results with basic stats would be more helpful to demonstrate the effectiveness of the proposed algorithm.

---

> ### Author Rebuttal · Authors · 2024-08-06
>
> **W1:** FedGH achieves FL collaboration across clients with heterogeneous local models by sharing a co-training homogeneous prediction header at the server. It can be categorized into the model split branch. However, sharing one part of the heterogeneous client model may result in insufficient generalization for the local complete model and overfitting for the local remaining part, leading to model performance degradation and also disclosing partial model structure privacy.
>
> ProxyFL enables clients with heterogeneous models to exchange knowledge by sharing a proxy homogeneous model, it belongs to mutual learning-based model-heterogeneous FL methods like FML, FedKD, and FedAPEN. These methods utilize mutual loss or distillation loss calculated by the distance between the predictions of the shared proxy homogeneous model and the heterogeneous client model to train two models alternatively. The two models only exchange limited knowledge through the loss, resulting in model performance bottlenecks.
>
> FedMRL adds a proxy homogeneous model shared by clients with heterogeneous models for federated learning. Its innovative contributions mainly include the following two points. Owing to these designs, FedMRL can perform good model performance while effectively protecting heterogeneous model structure privacy, compared with FedGH and ProxyFL.
>
> - **Adaptive Representation Fusion:** We designed a lightweight personalized representation projector to fuse the global generalized representation extracted by the shared global proxy homogeneous model and the local personalized representation extracted by the heterogeneous client model. The local projector and the two models are trained in an end-to-end manner, so the projector is adaptive to local non-IID data distribution, implementing personalized adaptive representation fusion.
> - **Multi-Perspective Matryoshka Representation Learning:** Based on the fused representation which includes both global generalized and local personalized feature information, we construct multi-dimension and multi-granularity Matryoshka representations and improve model learning capability through Matryoshka representation learning. After local model training, only the proxy homogeneous models are transmitted between the server and clients and complete heterogeneous client models are always stored within clients, protecting client model structure privacy.
>
> **W2:** The extra computational cost at clients is increased by additionally training a small proxy homogeneous model and a lightweight one-linear-layer representation projector. Since the additional two models constitute only about 9% of the entire model, the additional computational cost in our FedMRL is minimal. In comparison to baseline methods in the same category that utilize proxy models, such as FML, FedKD, and FedAPEN, FedMRL's additional computational cost is lower, specifically, 9.21MB and 6.49MB FLOPs in **Table X2**. This is primarily due to our method's reduction in representation dimension. Our experiments, as illustrated in Figure 4, demonstrate that FedMRL maintains efficient computation while achieving significant improvements in model accuracy compared to other baselines, as shown in Table 1. Considering the significant performance improvement, we believe the minimal additional computational cost is a worthwhile compromise.
>
> The qualitative analysis was illustrated in **Appendix D**, lines 494-500, and the quantitive analysis was reported in Section 5.2.4. Both the qualitative and quantitive analysis demonstrate that although FedMRL increases slight extra computational overheads in one communication round, the adaptive representation fusion and multi-perspective Matryoshka representation learning enhance model generalization and personalization, speeding up model convergence. Therefore, as shown in Figure 4, FedMRL needs fewer communication rounds to reach the specified target model accuracy than the best baseline, and it also consumes lower total computational costs for reaching the target accuracy due to faster model convergence. Hence, FedMRL is also computationally efficient.
>
> **W3:** FedMRL can be applied to FL clients with arbitrary model structures since it fuses representations with an adaptive representation projector with a dimension that is adjusted freely for clients.
>
> We supplement extra experiments on the CIFAR-100 dataset with 100 clients under more complicated model heterogeneity including ResNet-{4, 6, 8, 10, 18, 34, 50, 101, 152}, CNN, and a ViT model. The results in **Table X1** again validate the state-of-the-art performance of FedMRL. ‘-’ denotes failure to converge.
>
> **Q1:** Thanks for your valuable suggestions. We have tried our best to address the above concerns.
>
> **Q2:** Matryoshka representation learning has been substantiated to be an effective and efficient method to improve model learning capability. Inspired by it, we construct multi-dimension and multi-granularity Matryoshka representations processed by the global proxy homogeneous model’s prediction header and the local heterogeneous client model’s prediction header, respectively. The Matryoshka representation learning enables clients to learn the global generalized knowledge and local personalized knowledge from multiple perspectives, enhancing both model generalization and personalization and hence improving model performance.
>
> **Q3:** The optimization at the client is conducted in an end-to-end way.
>
> **Limitation:** Thanks for your advice. We conducted 3 trials of each experiment and only reported the average result. We will supplement the corresponding variances in the revised version.

---

> > ### Comment · Reviewer_NjYm · 2024-08-13
> >
> > Thanks for your reply. I will keep my score positive as it is. Thank you.

---

> > > ### Author Response · Authors · 2024-08-13
> > >
> > > Thank you so much for your positive feedback and continued support. We greatly appreciate your valuable comments!

---

### Official Review · Reviewer_DqWB · 2024-07-13

**Soundness:** 4
**Presentation:** 4
**Contribution:** 3
**Rating:** 6
**Confidence:** 5

**Summary:**

The paper proposed a FedMRL method for model-heterogeneous FL, which adapted Matryoshka Representation Learning to learn representations from multiple granularities.

**Strengths:**

1. The proposed method is a new way to tackle the heterogeneous challenge of federated learning.

2. The paper is well-organized and easy to follow.

3. Abundant experiments and theoretical analysis demonstrate the effectiveness of the proposed methods.

**Weaknesses:**

1. The novelty of the proposed solution needs a stronger justification. There are some representation fusion-based solutions to tackle heterogeneous federated learning challenges, for example, Federated Self-supervised Learning [12] and Federated Contrastive Learning [2].

[1] Weiming Zhuang, et al., DIVERGENCE-AWARE FEDERATED SELF-SUPERVISED LEARNING, ICLR 2022
[2] Qinbin Li, et al., Model-Contrastive Federated Learning, CVPR 2022
[3] Yue Tan, et al., Federated Learning from Pre-Trained Models: A Contrastive Learning Approach, NeurIPS 2022

2. The theoretical analysis shows no significant relevance to the Matryoshka Representation Learning.

**Questions:**

1. In Figure 1, the Matryoshka Representation is a key component for performance improvement. Does the Matryoshka Representation still capable of improving performance in a deeper CNN, or other neural architectures, e.g. ResNet, UNet and Transformer?

2. Are there any other Multi-Granularity Representation Learning methods other than Matryoshka Representation Learning?

---

> ### Author Rebuttal · Authors · 2024-08-06
>
> **W1:** We acknowledge the pointed related work heterogeneous federated learning. However, our approach, FedMRL, introduces significant innovations that differentiate it from existing methods. We appreciate the oppporutnity to highlight our novelty.
>
> **Frist, The referenced works do not fuse representation.** The referenced 3 FL contrastive learning methods use the representation distance of the shared global model and the local model as the training loss to enhance the generalization of the local model, which is only suitable for clients with homogeneous models since the server is required to aggregate them.
>
> **Second, Innovative Contributions in FedMRL**.  For representation learning in FedMRL, we propose two key innovative contributions:
> - **Innovative Representation Fusion:** We fuse the global generalized representations extracted by the proxy homogeneous model and the local personalized representations extracted by the heterogeneous client model through a training local personalized representation projector. The representation projector and the two models are simultaneously trained in an end-to-end manner, so the representation projector achieves a personalized representation fusion which is adapting to non-IID local data distributions.
> - **Multi-Perspective Representation Learning:** Based on the fused representation, we construct multi-dimension and multi-granularity Matryoshka representations. Each embedded dimension of Matryoshka representations is respectively processed by the proxy homogeneous model’s prediction head and the heterogeneous client model’s prediction head to output predictions and then compute loss with the ground-truth label, the loss sum is as the final loss to update all models in an end-to-end manner. In short, FedMRL innovatively utilizes multi-perspective Matryoshka representation learning to learn the global generalized feature and the local personalized feature from multiple perspectives, which is beneficial to improve model generalization and personalization simultaneously.
>
> Owing to these designs, FedMRL can be freely applied to more practical FL scenarios where clients may hold structure-heterogeneous models.
>
> **W2:** We would like to clarify that our theoretical analysis serves as a pivotal motivation for proposing suitable techniques. In our theoretical analysis, we derive the non-convex convergence rate of FedMRL based on a complete communication round. One key component impacting the overall convergence is the local training of the whole model (the proxy homogeneous model, the heterogeneous client model, and the local representation projector) at clients. The proposed Matryoshka Representation Learning method is known for better convergence and generalization in model training, as evidenced by [1]. This implies its ability to achieve better local model training. As shown in our Theorem 2, our convergence analysis indicates that the overall convergence benefits from improved local model training convergence. Therefore, we introduced Matryoshka Representation Learning in our framework to enhance this aspect. Additionally, our empirical results in Figure 3 confirm that FedMRL converges to higher model accuracy with a faster convergence speed compared to state-of-the-art baselines. This empirical evidence supports the theoretical claims and highlights the significant relevance of Matryoshka Representation Learning to the observed convergence behavior.
>
>
>
> **Q1:** Yes, the Matryoshka Representation can still improve model performance for deeper models. The core idea of the Matryoshka Representation is to add multiple prediction heads to process the embedded Matryoshka representations ranging from low-to-high dimensions and coarse-to-fine granularities to improve the learning capability of the encoder (i.e., feature extractor, model layers before the prediction head). This idea is inspired by the insight that people often first perceive a coarse outline of an observation objective and then carefully see the fine details, so multiple-perspective observations can enhance the understanding of one thing. For arbitrary shallow or deep models, all of them can construct Matryoshka Representations and append corresponding multiple prediction heads to improve model performance.
>
>
> **Q2:** Thank you for your insightful question. While there might be alternative solutions for Multi-Granularity Representation Learning, as far as we know, the Matryoshka Representation Learning method stands out as both effective and efficient, substantiated by extensive experiments.
>
> We chose the Matryoshka Representation Learning method for several reasons:
>
> - **Effectiveness:** The method has demonstrated significant improvements in model accuracy and performance through rigorous testing.
>
> - **Efficiency:** It allows for simultaneous training of both global and local models in an end-to-end manner, which is crucial for handling the non-IID data distributions in federated learning.
>
> FedMRL is the first to explore this method specifically in the FL domain, especially in the context of model-heterogeneous federated learning, addressing both generalization and personalization challenges. While we acknowledge that other methods may exist or emerge, we believe that Matryoshka Representation Learning currently provides a robust solution for the challenges at hand.
>
> [1] Matryoshka Representation Learning, NeurIPS 2022.

---

### Official Review · Reviewer_6tni · 2024-07-13

**Soundness:** 4
**Presentation:** 4
**Contribution:** 4
**Rating:** 7
**Confidence:** 4

**Summary:**

This paper focus on model heterogeneous  federated learning. Existing distillation-based learning results in limited knowledge transfer. In order to mitigate this challenge, the authors propose FedMRL. In FedMRL, each client trains an extra shared global auxiliary homogeneous small model such that the server can directly learn local data distribution from the auxiliary model. The authors provide theoretical convergence analysis for FedMRL. Experiments on benchmark datasets demonstrate the effectiveness of the proposed FedMRL.

**Strengths:**

* The idea of utilizing a small homogeneous model is novel and interesting.
* The writing is clear and easy to follow.
* Convergence analysis is provided.
* The experiments are extensive and can validate the effectiveness of the proposed method.

**Weaknesses:**

* FedMRL can add extra burden to computation power of client devices.
* The authors did not compare the theoretical convergence rate with traditional distillation based model heterogeneous federated learning.

**Questions:**

Please refer to weaknesses.

**Limitations:**

There is no potential negative societal impact of their work.

---

> ### Author Rebuttal · Authors · 2024-08-06
>
> **W1:** We would like to clarify that the additional computational cost in our proposed FedMRL is minimal. Specifically, it involves a small proxy homogeneous model and a one-linear-layer representation projector for the clients. Notably, the parameters of this additional component constitute only about 9% of the entire model. In comparison to baseline methods in the same category that utilize proxy models, such as FML, FedKD, and FedAPEN, FedMRL's additional computational cost is lower, specifically, 9.21MB and 6.49MB FLOPs in **Table X2**. This is primarily due to our method's reduction in representation dimension. Our experiments, as illustrated in Figure 4, demonstrate that FedMRL maintains efficient computation while achieving significant improvements in model accuracy compared to other baselines, as shown in Table 1. Considering the significant performance improvement, we believe the minimal additional computational cost is a worthwhile compromise.
>
> **W2:** We first restate our theoretical analysis. We prove the non-convex convergence rate through a complete communication round. To this end, we rely on the assumptions detailed in Section B in the Appendix introduce the error bounds associated with local training with hard loss (Lemma 1), and model aggregation (Lemma 2) and then derive the error bound of a once complete round of FL (Theorem 1). The convergence rate (Theorem 2) is based on training one complete round of FL by communicating and training small global homogenous proxy models and client heterogeneous models using hard loss. Note that, the baseline traditional distillation-based models (FD and FedProto) use additional knowledge distillation techniques that aggregate the output logits or representations of client models to generate the global logits or representations which are used in the next round’s local model training, such that additional loss and training schemes involved but do not fit in our theoretical framework due to their different model training and aggregation manners. Nevertheless, we have empirically demonstrated the better performance of FedMRL compared with the public data-free distillation-based methods FD and FedProto.

---

> > ### Comment · Reviewer_6tni · 2024-08-09
> >
> > Thanks for your response. I have raised my score.

---

> > > ### Author Response · Authors · 2024-08-10
> > >
> > > Thank you for your positive feedback and raising our score. Your valuable comments are greatly appreciated!

---

### Author Rebuttal · Authors · 2024-08-06

Please see Tables X1-X5 for rebuttal from the attached pdf file.

---

### Decision · Program_Chairs · 2024-09-25

**Decision:**

Accept (poster)

**Comment:**

The paper proposes a Matryoshka Representation Learning approach for tackling heterogenous FL. This is done by learning a small global auxiliary model. The authors show some convergence analysis to support the approach. The reviewers unanimously recommend acceptance.